# Mixed vine copulas as joint models of spike counts and local field potentials

**Arno Onken**
Istituto Italiano di Tecnologia
38068 Rovereto (TN), Italy
`arno.onken@iit.it`

**Stefano Panzeri**
Istituto Italiano di Tecnologia
38068 Rovereto (TN), Italy
`stefano.panzeri@iit.it`

## Abstract

Concurrent measurements of neural activity at multiple scales, sometimes performed with multimodal techniques, become increasingly important for studying brain function. However, statistical methods for their concurrent analysis are currently lacking. Here we introduce such techniques in a framework based on vine copulas with mixed margins to construct multivariate stochastic models. These models can describe detailed mixed interactions between discrete variables such as neural spike counts, and continuous variables such as local field potentials. We propose efficient methods for likelihood calculation, inference, sampling and mutual information estimation within this framework. We test our methods on simulated data and demonstrate applicability on mixed data generated by a biologically realistic neural network. Our methods hold the promise to considerably improve statistical analysis of neural data recorded simultaneously at different scales.

## 1 Introduction

The functions of the brain likely rely on the concerted interaction of its microscopic, mesoscopic and macroscopic systems. Concurrent recordings of signals at different scales, such as simultaneous measurements of field potential and single-cell spiking activity or other multimodal measures such as concurrent electrophysiological and fMRI measures, are leading to rapid advances in understanding brain dynamics [16]. Analysis of these concurrent data is complicated by the great difference in nature (e.g. discrete vs. continuous) and signal-to-noise ratio of each type of neural signal. To take full advantage of these data, flexible statistical models that take into account many variables with different statistics and their dependencies are needed. Recently, construction of multivariate statistical models based on the concept of copulas has attracted a lot of attention [9]. Intuitively, a copula represents a particular relationship between a set of random variables that, together with separate margin models of the individual elements can be used to construct a joint statistical model. This approach has become an indispensable tool in economics, finance and risk management in both theoretical and practical applications [9, 13, 11]. Yet, despite their promise, application to neuroscience has been limited [10, 14, 19]. The copula approach is general and, in principle, applicable to model mixed discrete and continuous statistics. Specific cases of mixed discrete and continuous copula-based models with parametric distributions were recently applied in clinical applications [24, 7]. Racine [17] proposed nonparametric mixed copula distributions based on kernel density estimators. In most studies, however, the elements of the copula-based multivariate distributions are all continuous [9, 13, 11]. A reason for this is that in the general case, likelihood calculation has exponential complexity in the number of discrete elements, limiting the usefulness of the models. In particular, these methods are impractical for likelihood-based estimation of information-theoretic quantities which requires many likelihood evaluations. Smith and Khaled [23] recently proposed a copula-based framework with quadratic complexity, but limited to fully discrete distributions. For valuable applications in neuroscience settings, however, we need a framework

that can overcome these limitations and cope with elements (i.e. number of neurons, activity sites) that have different statistical properties - some continuous and others discrete - while still allowing efficient likelihood calculation.

Here, we develop a framework to accomplish these goals by means of vine copulas with mixed discrete and continuous margins. We describe methods to make numeric model selection, parameter fitting and sampling scale efficiently with the number of elements and apply these methods to estimate information-theoretic quantities. To demonstrate our framework, we draw samples from mixed models and simulate mixed activity in a biologically realistic neural network. We then apply our methods to these data and show that our methods outperform corresponding mixed independent and fully continuous models.

## 2 Mixed vine copulas

Our goal is to construct multivariate distributions with arbitrary mixed margins and a wide range of possible dependence structures. To accomplish this goal, we apply an approach that individuates the margin part and the dependence part. The dependence is represented by a copula. Briefly, a copula is defined as a multivariate distribution function with support on the unit hypercube and uniform margins [13]. We will denote multivariate random variables by $\boldsymbol{X}$ with elements $X_i$. We denote the cumulative distribution function (CDF) of $\boldsymbol{X}$ by $F_{\boldsymbol{X}}$ with margin CDFs $F_i$. For consistency of notation, we will denote probability density functions as well as probability mass functions by $f_{\boldsymbol{X}}$ with margins $f_i$.

### 2.1 Mixed copula-based models

The great strength of copulas is their utility for constructing and decomposing multivariate distributions. Sklar's Theorem [21, 13] lays out the theoretical foundations for this. According to this theorem, every CDF $F_{\boldsymbol{X}}$ can be decomposed into margins $F_1, \ldots, F_d$ and a copula $C$ such that

$$F_{\boldsymbol{X}}(x_1, \ldots, x_d) = C(F_1(x_1), \ldots, F_d(x_d)) \tag{1}$$

and, conversely, margins $F_1, \ldots, F_d$, a copula $C$ and Eq. 1 can be used to construct a CDF $F_{\boldsymbol{X}}$. In this decomposition, $C$ is unique on the range of $\boldsymbol{X}$. Sklar's Theorem holds for mixed discrete and continuous distributions and thus provides a method to construct multivariate mixed distributions based on CDFs of copulas and margins. The important point here is that the approach yields a cumulative distribution function $F_{\boldsymbol{X}}$ of a multivariate random variable $\boldsymbol{X}$, not its likelihood $f_{\boldsymbol{X}}$ which we need for inference and other tasks (c.f. Section 2.5). Thus, we need to calculate the likelihood $f_{\boldsymbol{X}}$ based on the cumulative distribution function $F_{\boldsymbol{X}}$.

W.l.o.g., let $X_1, \ldots, X_n$ be discrete and $X_{n+1}, \ldots, X_d$ be continuous. By calculating the mixed derivative of Eq. 1, we obtain the probability density function of the mixed distribution of $\boldsymbol{X}$:

$$f_{\boldsymbol{X}}(x_1, \ldots, x_d) = \sum_{m_1=0,1} \cdots \sum_{m_n=0,1} (-1)^{m_1+\cdots+m_n}$$

$$\frac{\partial^{d-n} C}{\partial u_{n+1} \ldots \partial u_d}(F_1(x_1 - m_1), \ldots, F_n(x_n - m_n), F_{n+1}(x_{n+1}), \ldots, F_d(x_d)) \prod_{i=n+1}^{d} f_i(x_i). \tag{2}$$

Note that the number of terms in the sum grows exponentially with the number of discrete variables. In general, the exponential number of terms prevents us from a direct evaluation of this equation. Nevertheless, we will see in the next section that we need to calculate the probability density function for likelihood-based estimation of differential entropy and mutual information. Therefore, we need an efficient way to calculate the probability density function that is tractable for many discrete variables. We will introduce methods to accomplish this in Section 2.5.

### 2.2 Information estimation with copulas and mixed margins

For continuous as well as mixed multivariate distributions, differential entropy $h(\boldsymbol{X})$ is defined as $h(\boldsymbol{X}) = -\int f_{\boldsymbol{X}}(\boldsymbol{x}) \log_2 f_{\boldsymbol{X}}(\boldsymbol{x}) d\boldsymbol{x}$, where $f_{\boldsymbol{X}}$ is a multivariate density which can also have mixed margins like the one in Eq. 2 [6, 20]. With this, the mutual information $I(\boldsymbol{X}; \boldsymbol{Y})$ between two

multivariate random variables $\boldsymbol{X}$ and $\boldsymbol{Y}$ with potentially mixed margins is given by $I(\boldsymbol{X};\boldsymbol{Y}) = h(\boldsymbol{X}) + h(\boldsymbol{Y}) - h(\boldsymbol{X},\boldsymbol{Y})$, where $h(\boldsymbol{X},\boldsymbol{Y})$ is the joint differential entropy of the joint distribution $(\boldsymbol{X},\boldsymbol{Y})$ with joint density $f_{\boldsymbol{X},\boldsymbol{Y}}$ [6, 20]. For high dimensional distributions, evaluation of the integral over the support of $f_{\boldsymbol{X}}$ is unfeasible. However, we can estimate the differential entropy and thereby the mutual information by means of classical Monte Carlo (MC) estimation [18]. We express the entropy as an expectation over $f_{\boldsymbol{X}}$ and approximate the expectation by the empirical average by producing a large number of samples $\boldsymbol{x}_1, \ldots, \boldsymbol{x}_k$ from $\boldsymbol{X}$:

$$h(\boldsymbol{X}) = \mathbb{E}_{f_{\boldsymbol{X}}}\left[-\log_2 f_{\boldsymbol{X}}(\boldsymbol{X})\right] \approx \widehat{h_k} := -\frac{1}{k}\sum_{j=1}^{k} \log_2(f_{\boldsymbol{X}}(\boldsymbol{x}_j)) \qquad (3)$$

By the strong law of large numbers, $\widehat{h_k}$ converges almost surely to $h(\boldsymbol{X})$. Moreover, we can assess the convergence of $\widehat{h_k}$ by estimating the sample variance of $\widehat{h_k}$: $\operatorname{Var}\left[\widehat{h_k}\right] \approx \frac{1}{k+1}\sum_{j=1}^{k}\left(-\log_2(f_{\boldsymbol{X}}(\boldsymbol{x}_j)) - \widehat{h_k}\right)^2$. With this estimate, the term $\frac{\widehat{h_k} - h(\boldsymbol{X})}{\sqrt{\operatorname{Var}[\widehat{h_k}]}}$ is approximately standard normal distributed, allowing us to obtain confidence intervals for our differential entropy estimate [18]. This shows that there are two requisites for the MC procedure to estimate entropy and mutual information for a mixed distribution: 1) an efficient sampling procedure to produce samples $\boldsymbol{x}_j$ from $\boldsymbol{X}$, and 2) a tractable method for calculating the density $f_{\boldsymbol{X}}(\boldsymbol{x}_j)$. We will introduce the former in Section 2.4 and the latter in Section 2.5. In the next section we will describe a copula decomposition that makes these efficient methods possible.

## 2.3 Pair copula constructions

The number of available high-dimensional copula families is quite limited while there are an abundant number of bivariate copula families. The pair copula construction provides a flexible way to construct higher-dimensional copulas from bivariate copulas [1]. The idea of pair copula models is to factorize the multivariate distribution into conditional distributions and to describe these conditional distributions by means of bivariate copulas modeling dependence of two variables at a time. Special pair copula constructions, called regular vine copula structures, assume conditional independence between specific elements of the distribution, allowing us to circumvent the curse of dimensionality in likelihood evaluation and sampling. More specifically, a vine can be represented as a hierarchical set of trees where each node corresponds to a conditional distribution function and each edge corresponds to a pair copula. The nodes of the lowest tree are the unconditional distribution margins with empty conditioning sets. Each tree in the hierarchy incorporates additional variables into the conditioning sets by means of its pair copulas. The results of these couplings then form the nodes of the next tree in the hierarchy, thus extending the conditioning sets from tree to tree. Here we focus on the canonical vine or C-vine in which each tree in the hierarchy has a unique node that is connected to all other nodes [1].

In this section, $F(x_i | x_{j_1}, \ldots, x_{j_k})$ denotes the conditional cumulative distribution function of $X_i$ given $X_{j_1}, \ldots, X_{j_k}$. In the C-vine, the multivariate model likelihood is factorized as follows [1]:

$$f_{\boldsymbol{X}}(x_1, \ldots, x_d) = \prod_{k=1}^{d} f(x_k) \prod_{j=1}^{d-1} \prod_{i=1}^{d-j} c_{j,i+j|1,\ldots,j-1}(F(x_j | x_1, \ldots, x_{j-1}), F(x_{i+j} | x_1, \ldots, x_{j-1}))$$

$$(4)$$

The C-vine is a good choice if there are outstanding variables with important dependencies to many other variables [2]. Such situations are commonly encountered in electrophysiology recordings where the same electrode might record a local field potential (LFP, acting as the outstanding variable) and statistically dependent spikes from nearby neurons.

## 2.4 Sampling from mixed canonical vines

For a vine with mixed margins, we sample from the corresponding continuous vine and apply the inversion method with the inverse of the margin cumulative distribution function to obtain mixed discrete and continuous samples.

In the following, $\frac{\partial C}{\partial u_1}$ denotes the partial derivative of the copula $C$ with respect to its first argument and $\frac{\partial C}{\partial u_2}$ denotes the partial derivative with respect to the second argument. For mixed C-vine

sampling, we take the algorithm for sampling from a continuous C-vine copula with uniform margins [1] and extend it by means of the inversion method to attach arbitrary continuous and discrete margins. The algorithm requires $(d-2)(d-1)/2 + d$ cumulative distribution function evaluations:

1. Sample $w_1, \ldots, w_d$ i.i.d. uniform on $[0, 1]$.
2. $v_{1,1} = w_1$.
3. $x_1 = F_1^{-1}(v_{1,1})$.
4. For $i = 2, \ldots, d$:
   (a) $v_{i,1} = w_i$.
   (b) For $k = i-1, i-2, \ldots, 1 : v_{i,1} \leftarrow F_{i|1,\ldots,k}^{-1}(v_{i,1}, v_{k,k})$,
       where $F_{i|1,\ldots,k} = \frac{\partial C_{k,i|1,\ldots,k-1}}{\partial u_1}$.
   (c) $x_i = F_i^{-1}(v_{i,1})$.
   (d) If $i < d$ then for $j = 1, \ldots, i-1 : v_{i,j+1} \leftarrow F_{i|1,\ldots,j}(v_{i,j}, v_{j,j})$,
       where $F_{i|1,\ldots,j} = \frac{\partial C_{j,i|1,\ldots,j-1}}{\partial u_1}$.
5. The result is $x_1, \ldots, x_d$.

The algorithm has quadratic complexity and is thus applicable to estimate information-theoretic quantities following the scheme outlined in Section 2.2.

## 2.5 Tractable algorithm for calculating mixed canonical vine densities

Panagiotelis et al. [15] introduced an algorithm for calculating the likelihood of specific discrete pair-copula decompositions. Notably, this algorithm has quadratic complexity in the number of elements in the multivariate distribution. Here, we generalize this algorithm to the mixed margins case and apply it to the C-vine. We apply a dynamic programming approach and build the likelihood in a bottom up fashion from vine level $T_0$ to level $T_d$. The algorithm has quadratic complexity and computes the density of a C-vine with mixed discrete and continuous margins. We abbreviate $F_{i|A}^+ := F_{i|A}^c := P(X_i \leq x_i | X_A = x_A)$ and $F_{i|A}^- := P(X_i \leq x_i - 1 | X_A = x_A)$. We write $f_{i|A} := f(X_i = x_i | X_A = x_A)$ if $X_i$ is continuous and $f_{i|A} := P(X_i = x_i | X_A = x_A)$ if $X_i$ is discrete. Moreover $\forall a, b \in \{+, -, c\} : C_{i,j|A}^{ab} := C_{i,j|A}(F_{i|A}^a, F_{j|A}^b)$. $\frac{\partial C}{\partial u_1}$ is the partial derivative of the copula $C$ with respect to its first argument and $\frac{\partial C}{\partial u_2}$ is the partial derivative with respect to $C$'s second argument. Consequently, for $w \in \{u, v\}$ we write $\frac{\partial C_{i,j|A}^{ab}}{\partial w} := \frac{\partial C_{i,j|A}}{\partial w}(F_{i|A}^a, F_{j|A}^b)$.

1. Level $T_0$: For $i = 1, \ldots, d$: evaluate $f_i = F_i^+ - F_i^-$ if $X_i$ is discrete and $f_i = f_i(x_i)$ if $X_i$ is continuous.
2. Levels $T_1, T_2, \ldots, T_{d-1}$: For $t = 1, \ldots, d-1$ and $i = t+1, \ldots, d$: Let $I_t = \{1, \ldots, t\}$. Let $L_t = \{1, \ldots, t-1\}$ if $t > 1$, and $L_t = \emptyset$ if $t = 1$.
   (a)
   $$\text{Evaluate} \begin{cases} \forall a, b \in \{+, -\} : C_{t,i|L_t}^{ab} & \text{if } X_t \text{ and } X_i \text{ discrete,} \\ \forall a \in \{+, -\} : C_{t,i|L_t}^{ac} \text{ and } \frac{\partial C_{t,i|L_t}^{ac}}{\partial u_2} & \text{if } X_t \text{ discrete and } X_i \text{ continuous,} \\ \forall b \in \{+, -\} : C_{t,i|L_t}^{cb} \text{ and } \frac{\partial C_{t,i|L_t}^{cb}}{\partial u_1} & \text{if } X_t \text{ continuous and } X_i \text{ discrete,} \\ \frac{\partial C_{t,i|L_t}^{cc}}{\partial u_1}, \frac{\partial C_{t,i|L_t}^{c}}{\partial u_2} \text{ and } \frac{\partial^2 C_{t,i|L_t}^{c}}{\partial u_1 \partial u_2} & \text{if } X_t \text{ and } X_i \text{ continuous.} \end{cases}$$
   (5)
   (b) Evaluate
      • if $X_i$ discrete:
      $$f_{i|I_t} = F_{i|I_t}^+ - F_{i|I_t}^-,$$
      (6)
      where
      – if $X_t$ discrete:
      $$F_{i|I_t}^+ = \frac{C_{t,i|L_t}^{++} - C_{t,i|L_t}^{-+}}{f_{t|L_t}}, \qquad F_{i|I_t}^- = \frac{C_{t,i|L_t}^{+-} - C_{t,i|L_t}^{--}}{f_{t|L_t}}.$$
      (7)

– if $X_t$ continuous:

$$F^+_{i|I_t} = \frac{\partial C^{c+}_{t,i|L_t}}{\partial u_2}, \qquad F^-_{i|I_t} = \frac{\partial C^{c-}_{t,i|L_t}}{\partial u_2}. \tag{8}$$

• if $X_t$ discrete and $X_i$ continuous:

$$F^c_{i|I_t} = \frac{C^{+c}_{t,i|L_t} - C^{-c}_{t,i|L_t}}{f_{t|L_t}}, \; f_{i|I_t} = \frac{\partial F^c_{i|I_t}}{\partial x_i} = \left( \frac{\partial C^{+c}_{t,i|L_t}}{\partial u_1} - \frac{\partial C^{-c}_{t,i|L_t}}{\partial u_1} \right) \frac{f_{i|L_t}}{f_{t|L_t}}, \tag{9}$$

• if $X_t$ continuous and $X_i$ continuous:

$$F^c_{i|I_t} = \frac{\partial C^{cc}_{t,i|L_t}}{\partial u_2}, \qquad f_{i|I_t} = \frac{\partial F^c_{i|I_t}}{\partial x_i} = \frac{\partial^2 C^{cc}_{t,i|L_t}}{\partial u_1 \partial u_2} f_{i|L_t}, \tag{10}$$

3. The result is $f_{1,\ldots,d} = f_1 \prod_{i=2}^d f_{i|1,\ldots,i-1}$.

Like the sampling algorithm in Section 2.4, the likelihood algorithm has quadratic complexity and is thus applicable to estimate information-theoretic quantities following the scheme outlined in Section 2.2.

## 2.6 Inference

We can apply maximum likelihood methods to estimate model parameters, because we can directly calculate the full likelihood of the model - even for high dimensions - following the procedure outlined in Section 2.5. Let $\mathcal{L}(\boldsymbol{\theta}, \boldsymbol{\lambda}_1, \ldots, \boldsymbol{\lambda}_d) = \sum_{j=1}^k \log f_{\boldsymbol{X}}(\boldsymbol{x}_j; \boldsymbol{\theta}, \boldsymbol{\lambda}_1, \ldots, \boldsymbol{\lambda}_d)$ denote the log likelihood of the joint probability density function, where $\boldsymbol{\theta}$ denotes the parameters of the chosen copula family. We can now apply the so-called inference for margins (IFM) method to estimate the parameters [11]. The idea of this method is to break the joint optimization of all parameters up into smaller optimization problems. For $i = 1, \ldots, d$, let $\mathcal{L}_i(\boldsymbol{\lambda}_i) = \sum_{j=1}^k \log f_i(x_{i,j}; \boldsymbol{\lambda}_i)$ denote the sum of log likelihoods of the marginal distribution $f_i(x_{i,j}; \boldsymbol{\lambda}_i)$, where $\boldsymbol{\lambda}_1, \ldots, \boldsymbol{\lambda}_d$ are the parameters of the chosen family of margins. The method proceeds in two steps. In the first step, the margin likelihoods are maximized separately: $\forall i = 1, \ldots, d : \widehat{\boldsymbol{\lambda}_i} = \underset{\boldsymbol{\lambda}_i}{\operatorname{argmax}}\{\mathcal{L}_i(\boldsymbol{\lambda}_i)\}$. In the second step, the full likelihood is maximized given the estimated margin parameters as $\widehat{\boldsymbol{\theta}} = \underset{\boldsymbol{\theta}}{\operatorname{argmax}}\{\mathcal{L}(\boldsymbol{\theta}, \widehat{\boldsymbol{\lambda}_1}, \ldots, \widehat{\boldsymbol{\lambda}_d})\}$.

Each of the individual optimization problems can be solved by means of a general multivariate optimization algorithm such as the trust-region-reflective algorithm [4]. Joe and Xu [11] showed that the IFM estimator is asymptotically efficient. The method is particularly attractive if the ratio of margin parameters to copula parameters is big. If the number of copula parameters is too big to be estimated in a single joint optimization, then the complexity of the copula model can be reduced by truncating the vine tree of the C-vine (truncated vine [1]). This corresponds to an independence assumption for higher vine levels and the validity of this simplification should be confirmed [22]. The families of margin and copula distributions can be selected using the Akaike information criterion (AIC) [3]: Each combination of family selections is scored by means of its AIC value and then the best combination is chosen.

## 3 Validation on artificial data

We validated our framework by sampling from mixed vine-based models of different dimensionality and by evaluating performance of various alternative models. Fig. 1 illustrates a 3-dimensional example vine-based model with two continuous margins and one discrete margin. In the top row, we show the probability density functions of the 2-dimensional margins obtained by integrating over one margin each. One can appreciate the mixed distribution from the step-wise changes in probability density in margin 2 and the smooth changes in margins 1 and 3. The bottom row shows scatter plots of 3-dimensional samples projected onto each pair of margins. The distributions of samples nicely reflect the corresponding densities.

We drew samples from this and other mixed vine distributions and fitted various models to these samples. For model selection, we used normal and gamma distributions as options for continuous

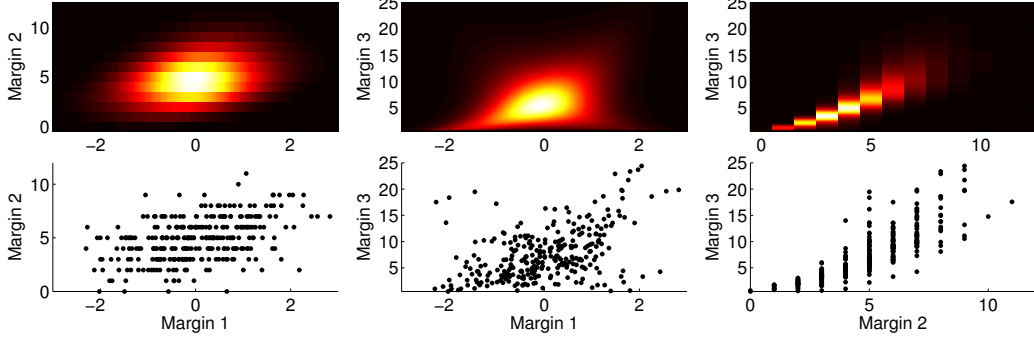

Figure 1: Characteristics of a 3D mixed vine example. Margin 1 is standard normal distributed, margin 2 is Poisson distributed with mean 5 and margin 3 is gamma distributed with shape 2 and scale 4. The pairwise copulas are Gaussian with parameter 0.5, Student with correlation 0.5 and 2 degrees of freedom and Clayton with parameter 5 for margin pairs (1,2), (1,3) and (2,3) respectively. Top row: Probability density functions of 2D margins. The lighter the color the higher is the density. Bottom row: 2D margin scatter plots of 300 samples.

margins, Poisson, binomial and negative binomial distributions as options for discrete margins, and Gaussian, Student, Clayton and rotated $(90°, 180°, 270°)$ Clayton copula families as options for pair copula constructions. To quantify the gain of using a vine-based mixed model instead of a mixed independent model, we drew samples from the vine-based mixed model and calculated the cross-validated likelihood ratio (LR) statistic for nested models as $D = 2(\log(\mathcal{L}_{\text{vine}}) - \log(\mathcal{L}_{\text{ind}}))$, where $\mathcal{L}_{\text{vine}}$ denotes the likelihood of separate test-set samples under the vine-based model and $\mathcal{L}_{\text{ind}}$ denotes the likelihood of the samples under the corresponding independent model.

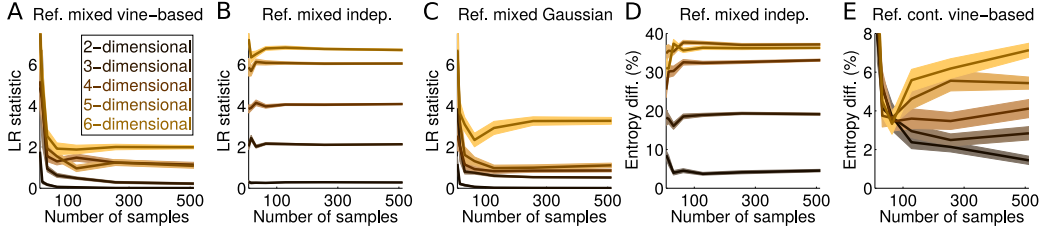

Figure 2: Model fit and entropy of simulated vine samples. Ground truth models are mixed vines of different dimensionality (range 2 to 6 shown as dark brown to light brown lines) with margins and copulas up to the respective dimension. Margins 1 to 3 and associated pairwise copulas are the same as in Fig. 1. Margin 4 is binomial distributed with $N = 6$ and $p = 0.4$, margin 5 is negative binomial distributed with $N = 6$ and $p = 0.4$ and margin 6 is standard normal distributed. The pairwise copulas are Clayton survival, independent and Clayton rotated $90°$ for margin pairs (1,4), (2,4) and (3,4) respectively, and Clayton rotated $270°$, independent, Gaussian with parameter 0.5 and independent for margin pairs (1,5), (2,5), (3,5) and (4,5) respectively and independent, independent, Gaussian with parameter 0.5, independent and Student with parameters 0.5 and 2 for margin pairs (1,6), (2,6), (3,6), (4,6) and (5,6) respectively and with parameter 5 for all Clayton based copulas. (A-C) Cross-validated LR statistic between the ground truth model and the mixed vine-based model (A), independent model (B) or mixed Gaussian model (C). (D,E) Normalized entropy difference between the ground truth model and the independent model (D) or fully continuous vine-based model (E). Lines denote averages over 30 repetitions as a function of the number of samples. Shaded areas denote standard error.

Fig. 2A shows the LR statistic between the ground truth and the best-fitting mixed vine-based model as a function of the number of samples for different dimensionality. The statistics were low in all cases but increased with increasing dimensionality. The gain as quantified by the LR statistic of using the full mixed vine-based model instead of the independent model, on the other hand, was moderate for the bivariate model ($D < 0.5$) while being substantial for the 6-dimensional model ($D \approx 7$). Wilks' LR test on non-cross-validated data was highly significant whenever we used at least 32 samples

($p < 0.01$). We also evaluated the fit of the multivariate Gaussian copula with mixed margins which is nested in our mixed vine-based models and obtained by restricting all pairwise copula families to be Gaussian. The LR statistics indicated substantially better fit than for the independent model but the statistics were below those of the mixed vine-based model for most tested dimensions (Fig. 2C).

Unfortunately, a vine-based mixed model and the corresponding best-fitting fully continuous vine-based model are not directly comparable in this way due to the different weighting of discrete and continuous elements (i.e. mass vs. density). Nevertheless, in an actual application it is easy to determine which margins are discrete and which margins are continuous. Appropriate discrete or continuous margins can therefore be selected easily. To extend our comparison to fully continuous vine-based models, we estimated entropies of the mixed vine-based model, the corresponding independent model and of the best-fitting fully continuous model. We calculated the entropy differences between these models and normalized with the entropy of the mixed vine-based model. Fig. 2D shows the normalized entropy difference between the mixed vine-based model and the independent model. The relative results are similar to those of the likelihood ratio statistic (Fig. 2B) suggesting that in this case the entropy comparison is indicative of the performance gain. In Fig. 2E, we plot the normalized entropy difference between the mixed vine-based model and the best-fitting fully continuous model. Overall, the normalized differences of these models were smaller than for the independent model. Similarly to the independent model, though, we found increasing differences for increasing dimensionality of the models. All in all, our results suggest that our framework can yield substantial advantages in terms of goodness of fit and in terms of estimated entropy in particular for high-dimensional problems.

## 4  Application to simulated network activity

To evaluate our framework in a typical neuroscience setting, we applied our mixed vine-based model to a biologically realistic neural network model. We simulated network activity with the Virtual Electrode Recording Tool for EXtracellular potentials (VERTEX) [25] with network parameters as in VERTEX tutorial 2. Briefly, the model contained a total of 5000 neurons with 85% of those cell models representing layer 2/3 pyramidal neurons and 15% representing basket interneurons. The spiking dynamics followed an adaptive exponential model. To simulate two different stimulus conditions, we used random input currents with different means. We presented each stimulus condition an equal number of times (corresponding to $1/2$ probability of occurrence of either stimulus). The network generated network oscillations in both conditions. To simulate a typical recording situation, we recorded LFPs with two randomly placed electrodes and collected spike counts from the four neurons closest to those electrodes. For each input condition, we ran the network 128 times and collected one 6-dimensional mixed vector with the LFPs (continuous) and spike counts (discrete) collected in a $100\,\mathrm{ms}$ interval from each network run. We then fitted the full mixed vine-based model, the mixed independent model and the fully continuous vine-based model to these data. Importantly, we fitted separate models for each stimulus condition and varied the number of samples per stimulus condition between 8 and 128. This allowed us to estimate mutual information following the procedure outlined in Section 2.2.

Similarly to Figs. 2B,C, Fig. 3A depicts the LR statistic between the best-fitting mixed vine-based model and the corresponding independent model or mixed Gaussian model. We found relatively small statistics for all sample sizes ($D < 1$). Nevertheless, Wilks' LR test indicated highly significant improvement whenever we used at least 64 samples ($p < 0.01$). To evaluate the importance of the mixed vine-based model when performing an information-theoretic analysis of the network activity, we estimated mutual information between the modeled network activity (LFP and spike counts) and the two stimulus conditions. Fig. 3B shows mutual information estimates that we obtained based on the mixed independent, mixed Gaussian, continuous vine-based and mixed vine-based models. The mixed Gaussian model yielded information estimates that were close to those of the mixed-vine based models. Estimates based on the independent model and fully continuous model, on the other hand, were both substantially different (overestimating and underestimating information, respectively) from estimates that we obtained from the mixed vine-based model. The latter model is the most faithful one with the most accurate information estimates. The overestimation of the independent model suggests that spike counts and LFPs carry partly redundant information. The big differences in information estimates further indicate that it can be important to take mixed margins and dependencies into account for estimating mutual information, even if the LR statistic is low.

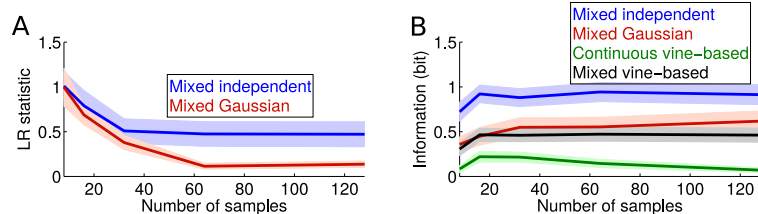

Figure 3: Analysis of simulated neural network activity obtained from the VERTEX tool [25]. Data samples are formed by the average LFP within $200 - 300 \, \mathrm{ms}$ after simulation onset from two randomly chosen electrodes and spike counts from the four neurons in closest proximity to those electrodes. One simulation run provided one sample only. The network was simulated with two different input conditions: Input currents following an Ornstein-Uhlenbeck process had a mean value of $330 \, \mathrm{pA}$ for the excitatory population and $190 \, \mathrm{pA}$ for the inhibitory population in condition 1, and $300 \, \mathrm{pA}$ for the excitatory population and $40 \, \mathrm{pA}$ for the inhibitory population in condition 2. In both conditions, standard deviation was $90 \, \mathrm{pA}$ for the excitatory population and $50 \, \mathrm{pA}$ for the inhibitory population. (A) LR statistic between the best-fitting mixed vine-based model and the best-fitting mixed independent model (blue) or mixed Gaussian model (red) as a function of the number of samples (i.e. number of simulations in each condition) averaged over stimulus conditions. (B) Mutual information between the neural activity and the two input conditions estimated from the mixed independent model (blue), mixed Gaussian model (red), continuous vine-based model (green) or mixed vine-based model (black) as a function of the number of samples. Lines denote averages over 30 repetitions. Shaded areas denote standard error.

## 5   Discussion

We developed a complete framework based on vine copulas for modeling multivariate data that are partly discrete and partly continuous. Our framework includes methods for sampling, likelihood calculation and inference. We combined these procedures to estimate entropy and mutual information by means of MC integration. In particular, our methods provide the possibility to construct joint statistical models of LFPs and spike counts. In a biologically realistic network simulation we demonstrated that our mixed vine-based model provides a fit that is better than that of the corresponding independent model and showed that mutual information estimates of fully continuous and mixed independent models can strongly differ even if the likelihood ratio statistic suggests otherwise. For LFP and spike count data, a mixed model with detailed dependence structures can make full use of all available statistical data. This also makes it possible to construct optimal Bayesian decoders for inferring the presented stimulus from both LFPs and spike counts. Moreover, our model provides the possibility to investigate the statistical dependencies between LFPs and spike counts. Contrary to other analysis methods for analyzing mixed LFPs and spiking [12, 5] our framework follows a purely data-driven approach. Even high-dimensional distributions can be fitted, because all inference operations have quadratic complexity. However, entropy and MI estimation can be problematic, because MC integration can become unfeasible for very high-dimensional problems. One possible remedy is to use our models for maximum likelihood decoding and then estimate information based on decoding performance [8]. We note that our models are based on pair-constructions and thus cannot model arbitrary higher-order dependencies. We stress, however, that higher-order correlations do occur in the vine tree and depend on both the vine-tree selection and the copula families. Thus, selecting the right vine-tree and copula families can - to a limited extent - account for higher-order correlations. In general, however, limited sample numbers make it difficult to reliably estimate higher-order correlations in real neuroscience applications. The parametric nature of our model framework also makes it possible to introduce dependencies on external variables. Directions for future research include applications to experimentally recorded data and detailed evaluation of observed dependency structures.

**Acknowledgments.**   This work was supported by the European Commission's Horizon 2020 Programme (H2020-MSCA-IF-2014) under grant agreement number 659227 ("STOMMAC").

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
