[Supplementary Material]

# Mixed vine copulas as joint models
# of spike counts and local field potentials

Barcelona, Spain.

## Supplementary information

## 1   Formal copula definition

Intuitively, a copula is a distribution function on the unit hypercube with uniform margins. In what follows, we give the formal definition of a multivariate copula [5]:

**Definition 1.** *A $d$-copula is a function $C : [0,1]^d \longrightarrow [0,1]$ such that $\forall \boldsymbol{u} \in [0,1]^d$:*

1. *$C(\boldsymbol{u}) = 0$ if at least one coordinate of $\boldsymbol{u}$ is $0$.*

2. *$C(\boldsymbol{u}) = u_k$ if all coordinates of $\boldsymbol{u}$ are $1$ except $u_k$.*

3. *Let $V_C([\boldsymbol{u}, \boldsymbol{v}]) = \sum_{i_1=1}^{2} \cdots \sum_{i_d=1}^{2} (-1)^{i_1+\cdots+i_d} C(g_{1,i_1}, \ldots, g_{d,i_d}), g_{j,1} = u_j, g_{j,2} = v_j,$
   then $V_C([\boldsymbol{u}, \boldsymbol{v}]) \geq 0$ for all $\boldsymbol{v} \in [0,1]^d$ with $\boldsymbol{u} \leq \boldsymbol{v}$.*

The second property implies uniform margins, whereas the first and third properties ensure that $C$ is a proper CDF on the unit hypercube. In particular, the third property ensures that every hypercube contains non-negative mass.

The great strength of copulas is their utility for constructing multivariate distributions. They can be used to couple arbitrary marginal CDFs to form a joint CDF. Sklar's Theorem [6, 5] lays out the theoretical foundations for this construction:

**Theorem 2.** *Let $F_{\boldsymbol{X}}$ be a $d$-dimensional cumulative distribution function with margins $F_1, \ldots, F_d$. Then there exists a $d$-copula $C$ such that $\forall \boldsymbol{x} \in Domain(F_{\boldsymbol{X}})$:*

$$F_{\boldsymbol{X}}(x_1, \ldots, x_d) = C(F_1(x_1), \ldots, F_d(x_d)). \tag{S1}$$

*$C$ is unique, if $X_1, \ldots, X_d$ are all continuous, and unique on $Range(F_1) \times \cdots \times Range(F_d)$, if $X_1, \ldots, X_d$ are discrete. Conversely, if $C$ is a $d$-copula and $F_1, \ldots, F_d$ are CDFs, then the function $F_{\boldsymbol{X}}$ defined by $F_{\boldsymbol{X}}(x_1, \ldots, x_d) = C(F_1(x_1), \ldots, F_d(x_d))$ is a $d$-dimensional CDF with margins $F_1, \ldots, F_d$.*

Sklar's Theorem allows us to construct multivariate distributions by attaching margin CDFs to copulas and to decompose multivariate CDFs into its margins and a copula.

## 2   Copula-based mixed likelihoods

The construction of a copula-based model yields a CDF whereas many applications (e.g. inference, entropy estimation) require the likelihood function. Hence, we need to calculate the likelihood based on the CDF.

Let $\boldsymbol{X}$ denote a multivariate distribution with CDF $F_{\boldsymbol{X}}(x_1,\ldots,x_d) = C(F_1(x_1),\ldots,F_d(x_d))$ where $C$ is a copula and $F_1$, ..., $F_d$ denote the margin CDFs. If $\boldsymbol{X}$ is continuous, then we can easily evaluate $f_{\boldsymbol{X}}$:

$$f_{\boldsymbol{X}}(x_1,\ldots,x_d) = \frac{\partial^d C}{\partial u_1 \ldots \partial u_d}(F_1(x_1),\ldots,F_d(x_d))\prod_{i=1}^{d} f_i(x_i). \tag{S2}$$

If $\boldsymbol{X}$ is discrete, however, then $f_{\boldsymbol{X}}$ is a probability mass function (PMF). To calculate the PMF, let $A = \{X_1 \leq x_1,\ldots,X_d \leq x_d\}$ and $A_i = \{X_1 \leq x_1,\ldots,X_d \leq x_d, X_i \leq x_i - 1\}$, $i \in \{1,\ldots,d\}$. The probability $P$ of a $d$-tuple of discrete values $\boldsymbol{x} = (x_1,\ldots,x_d)$ can now be evaluated using only the cumulative distribution function $F_{\boldsymbol{X}}$ by applying the principle of inclusion-exclusion [4]:

$$f_{\boldsymbol{X}}(x_1,\ldots,x_d) = P\left(A \setminus \bigcup_{i=1}^{d} A_i\right) = P(A) - \sum_{k=1}^{d}(-1)^{k-1}\sum_{\substack{I\subseteq\{1,\ldots,d\},\\|I|=k}} P\left(\bigcap_{i\in I} A_i\right)$$

$$= F_{\boldsymbol{X}}(\boldsymbol{x}) - \sum_{k=1}^{d}(-1)^{k-1}\sum_{\substack{\boldsymbol{m}\in\{0,1\}^d,\\\sum m_i=k}} F_{\boldsymbol{X}}(x_1 - m_1,\ldots,x_d - m_d) \tag{S3}$$

$$= \sum_{m_1=0,1}\cdots\sum_{m_d=0,1}(-1)^{m_1+\cdots+m_d} C(F_1(x_1 - m_1),\ldots,F_d(x_d - m_d)).$$

Similarly, if the margins are mixed, then we can combine the approaches for continuous and discrete likelihood calculation to obtain the probability density of the mixed distribution. W.l.o.g., let $X_1,\ldots,X_n$ be discrete and $X_{n+1},\ldots,X_d$ be continuous. Combining Eqs. S2 and S3, we obtain the probability density function of the mixed distribution of $\boldsymbol{X}$:

$$
\begin{aligned}
f_{\boldsymbol{X}}(x_1,\ldots,x_d) &= \frac{\partial^{d-n}}{\partial u_{n+1}\ldots\partial u_d}\Bigg[\sum_{m_1=0,1}\cdots\sum_{m_n=0,1}(-1)^{m_1+\cdots+m_n} \\
&\quad \cdot C(F_1(x_1 - m_1),\ldots,F_n(x_n - m_n),F_{n+1}(x_{n+1}),\ldots,F_d(x_d))\Bigg] \\
&= \sum_{m_1=0,1}\cdots\sum_{m_n=0,1}(-1)^{m_1+\cdots+m_n}\frac{\partial^{d-n}C}{\partial u_{n+1}\ldots\partial u_d} \\
&\quad (F_1(x_1 - m_1),\ldots,F_n(x_n - m_n),F_{n+1}(x_{n+1}),\ldots,F_d(x_d))\prod_{i=n+1}^{d} f_i(x_i).
\end{aligned}
\tag{S4}
$$

This equation has exponential complexity with the number of discrete elements.

## 3 Canonical vine illustration

The canonical vine or C-vine is as a pair copula construction that decomposes a multivariate density into a product of particular conditional distribution functions [2]. Each pair copula construction can be represented as a nested set of tree structures $\{T_1,\ldots,T_{d-1}\}$. In these trees, each edge corresponds to a pair copula density. The nodes in $T_1$ are the distribution functions of margins $1,\ldots,d$. The nodes in

Figure S1: Example C-vine with 4 elements. (A) Illustration of conditional elements and their relations. Each tree level adds one element to the conditioning set. (B) Corresponding nested trees $T_1, T_2$ and $T_3$. The nodes denote (conditional) cumulative distribution functions whereas the edges denote pair copulas that are used to construct the nodes of the next tree.

$T_2, \ldots, T_{d-1}$ are conditional distribution functions that stem from the edges of the previous tree: edges in tree $T_i$ become nodes in tree $T_{i+1}$. In the C-vine, each tree $T_i$ has a particular node that is connected to all other nodes in the tree.

The structure of a C-vine with 4 elements is illustrated in Fig. S1. The construction starts with the margins $1, \ldots, 4$ (level one, Fig. S1A top row). Each of the following levels then builds consecutive distributions of conditional elements: each level has an element that is connected to all other elements. In the illustration (Fig. S1), this is element 1 in level one, element 2 in level two and so forth. These elements are successively added to the conditioning sets of the following levels until all elements but one belong to the conditioning set (level four, Fig. S1A bottom row).

The construction of this C-vine is represented more accurately as a set of nested trees $\{T_1, \ldots, T_3\}$ (Fig. S1B). The nodes in tree $T_1$ are the marginal CDFs $1, \ldots, 4$. In this tree, element 1 is the one that is connected to all other elements by means of three copulas $C_{12}, C_{13}$ and $C_{14}$ which yield conditional CDFs $F_{2|1}, F_{3|1}$ and $F_{4|1}$, respectively. These conditional CDFs then form the nodes of tree $T_2$ and so forth.

Formally, the C-vine decomposition of a density $f_{\boldsymbol{X}}$ takes the following form [2]:

$$f_{\boldsymbol{X}}(x_1, \ldots, x_d) = \prod_{k=1}^{d} f(x_k) \prod_{j=1}^{d-1} \prod_{i=1}^{d-j} c_{j,i+j|1,\ldots,j-1}(F(x_j|x_1, \ldots, x_{j-1}), F(x_{i+j}|x_1, \ldots, x_{j-1})), \qquad (S5)$$

where $F(x_i|x_{j_1}, \ldots, x_{j_k})$ denotes the conditional cumulative distribution function of $X_i$ given $X_{j_1}, \ldots, X_{j_k}$.

# 4 Bivariate copula families for pair copula constructions

Here we propose copula families for modeling diverse dependency structures. The list partly follows [2] but our list contains notable differences to make the families more suitable for efficient likelihood calculation. For modeling upper tail dependence, we propose the survival transformation of the lower tail dependence Clayton copula instead of the Gumbel copula. The Gumbel copula would require numerical inversion of its conditional cumulative distribution function. The survival Clayton copula, on the other hand, can be treated fully analytically and thus does not require any numerical inversion. This choice has the additional advantage of having more consistent dependency structures: the shape of the survival Clayton copula is that of the mirrored original Clayton copula whereas the Gumbel copula has a different shape. We also propose to use the partial survival transformation with respect to only one element of the Clayton copula to obtain negative dependencies with tail dependence. With this procedure we have a versatile procedure for obtaining heavy tail dependence in all four corners of the bivariate distribution.

In the following, for each of the copula families, we list the cumulative distribution function, the density, the conditional cumulative distribution function and the inverse of the conditional cumulative distribution function. These are all the functions that are required for applying inference, sampling and likelihood calculations as proposed in the preceding sections. For each of the families, if $\theta = 0$ then the copula is the independence copula.

## 4.1 Bivariate Gaussian copula

The Gaussian copula is the dependence structure that is derived from the multivariate normal distribution. It is symmetric and does not have heavy tails.
**Cumulative distribution function:**

$$C(u_i, u_2; \theta) = \Phi_\theta \left( \Phi^{-1}(u_1), \Phi^{-1}(u_2) \right) = \int_{-\infty}^{u_1} \int_{-\infty}^{u_2} c(s, t; \theta) ds dt, \tag{S6}$$

where $\Phi$ is the cumulative distribution function of the standard normal distribution and $\Phi_\theta$ is the cumulative distribution function of the bivariate standard normal distribution function with correlation parameter $\theta$.
**Density:**

$$\frac{\partial^2 C(u_1, u_2; \theta)}{\partial u_1 \partial u_2} = c(u_1, u_2; \theta) \tag{S7}$$

$$= \frac{1}{\sqrt{1 - \theta^2}} \cdot \exp \left( \frac{2\theta \Phi^{-1}(u_1)\Phi^{-1}(u_2) - \theta^2((\Phi^{-1}(u_1))^2 + (\Phi^{-1}(u_2))^2)}{2(1 - \theta^2)} \right)$$

**Conditional cumulative distribution function:**

$$C_{1|2}(u_1|u_2; \theta) = \frac{\partial C(u_1, u_2; \theta)}{\partial u_2} = \Phi \left( \frac{\Phi^{-1}(u_1) - \theta \Phi^{-1}(u_2)}{\sqrt{1 - \theta^2}} \right) \tag{S8}$$

**Inverse of conditional cumulative distribution function with respect to $u_1$:**

$$C_{1|2}^{-1}(u_1|u_2; \theta) = \Phi \left( \Phi^{-1}(u_1)\sqrt{1 - \theta^2} + \theta \Phi^{-1}(u_2) \right) \tag{S9}$$

## 4.2   Bivariate Student copula

The Student copula is a symmetric copula like the Gaussian copula but with heavy tails. It is derived from Student's $t$ distribution.

**Cumulative distribution function:**

$$C(u_1, u_2; \boldsymbol{\theta}) = T_{\boldsymbol{\theta}}\left(T_{\theta_2}^{-1}(u_1), T_{\theta_2}^{-1}(u_2)\right) = \int_{-\infty}^{u_1}\int_{-\infty}^{u_2} c(s,t;\theta)dsdt, \tag{S10}$$

where $T_{\theta_2}$ is the cumulative distribution function of the standard Student's $t$ distribution and $T_{\boldsymbol{\theta}}$ is the cumulative distribution function of the bivariate standard Student's $t$ distribution function with correlation parameter $\theta_1$ and degrees of freedom $\theta_2$.

**Density:**

$$\frac{\partial^2 C(u_1, u_2; \boldsymbol{\theta})}{\partial u_1 \partial u_2} = c(u_1, u_2; \boldsymbol{\theta}) \tag{S11}$$

$$= \frac{\Gamma((\theta_2+2)/2)}{\Gamma(\theta_2/2)\pi\theta_2\sqrt{1-\theta_1^2}t_{\theta_2}(T_{\theta_2}^{-1}(u_1))t_{\theta_2}(T_{\theta_2}^{-1}(u_2))}$$

$$\cdot\left(1 + \frac{(T_{\theta_2}^{-1}(u_1))^2 - 2\theta_1 T_{\theta_2}^{-1}(u_1)T_{\theta_2}^{-1}(u_2) + (T_{\theta_2}^{-1}(u_2))^2}{\theta_2(1-\theta_1^2)}\right)^{-(\theta_2+2)/2},$$

where $t_{\theta_2}$ is the probability density function of the standard Student's $t$ distribution.

**Conditional cumulative distribution function:**

$$C_{1|2}(u_1|u_2; \boldsymbol{\theta}) = \frac{\partial C(u_1, u_2; \boldsymbol{\theta})}{\partial u_2} = T_{\theta_2+1}\left(\sqrt{\frac{\theta_2+1}{\theta_2+(T_{\theta_2}^{-1}(u_2))^2}}\frac{T_{\theta_2}^{-1}(u_1) - \theta_1 T_{\theta_2}^{-1}(u_2)}{\sqrt{1-\theta_1^2}}\right) \tag{S12}$$

**Inverse of conditional cumulative distribution function with respect to $u_1$:**

$$C_{1|2}^{-1}(u_1|u_2; \boldsymbol{\theta}) = T_{\theta_2}\left(\sqrt{\frac{(1-\theta_1^2)(\theta_2+(T_{\theta_2}^{-1}(u_2))^2)}{\theta_2+1}}T_{\theta_2+1}^{-1}(u_1) + \theta_1 T_{\theta_2}^{-1}(u_2)\right) \tag{S13}$$

## 4.3   Bivariate Clayton copula and survival Clayton copulas

We propose to use the Clayton copula for both lower and upper tail dependencies. The cumulative distribution function, the density, the conditional cumulative distribution function and its inverse can all be treated analytically and thus are applicable for efficient likelihood calculation. We apply survival transformations to switch from the original lower tail dependence to upper tail dependence and to negative dependence. In this section, we always set $\theta \in (0, \infty)$[1].

### 4.3.1 Original Clayton copula

This copula has lower tail dependence.
**Cumulative distribution function:**

$$C_{\text{Clayton}}(u_1, u_2; \theta) = \left( \max \left\{ u_1^{-\theta} + u_2^{-\theta} - 1, 0 \right\} \right)^{-1/\theta} \tag{S14}$$

**Density:**

$$\frac{\partial^2 C(u_1, u_2; \theta)}{\partial u_1 \partial u_2} = (1 + \theta)(u_1 u_2)^{-1-\theta} \left( u_1^{-\theta} + u_2^{-\theta} - 1 \right)^{-1/\theta - 2} \tag{S15}$$

**Conditional cumulative distribution function:**

$$C_{1|2}(u_1|u_2; \theta) = \frac{\partial C(u_1, u_2; \theta)}{\partial u_2} = \left( \max \left\{ u_2^{-(1+\theta)}(u_1^{-\theta} + u_2^{-\theta} - 1)^{-(1+1/\theta)}, 0 \right\} \right) \tag{S16}$$

**Inverse of conditional cumulative distribution function with respect to $u_1$:**

$$C_{1|2}^{-1}(u_1|u_2; \theta) = (1 - u_2^{-\theta} + (u_1 u_2^{1+\theta})^{-\theta/(\theta+1)})^{-1/\theta} \tag{S17}$$

### 4.3.2 Survival Clayton copula

This copula has upper tail dependence.
**Cumulative distribution function:**

$$
\begin{aligned}
C(u_1, u_2; \theta) &= u_1 + u_2 - 1 + C_{\text{Clayton}}(1 - u_1, 1 - u_2; \theta) \\
&= u_1 + u_2 - 1 + \left( \max \left\{ (1 - u_1)^{-\theta} + (1 - u_2)^{-\theta} - 1, 0 \right\} \right)^{-1/\theta}
\end{aligned}
\tag{S18}
$$

**Density:**

$$\frac{\partial^2 C(u_1, u_2; \theta)}{\partial u_1 \partial u_2} = (1 + \theta)((1 - u_1)(1 - u_2))^{-1-\theta} \left( (1 - u_1)^{-\theta} + (1 - u_2)^{-\theta} - 1 \right)^{-1/\theta - 2} \tag{S19}$$

**Conditional cumulative distribution function:**

$$C_{1|2}(u_1|u_2; \theta) = \frac{\partial C(u_1, u_2; \theta)}{\partial u_2} = 1 - \left( \max \left\{ (1 - u_2)^{-(1+\theta)}((1 - u_1)^{-\theta} + (1 - u_2)^{-\theta} - 1)^{-(1+1/\theta)}, 0 \right\} \right) \tag{S20}$$

**Inverse of conditional cumulative distribution function with respect to $u_1$:**

$$C_{1|2}^{-1}(u_1|u_2; \theta) = 1 - (1 - (1 - u_2)^{-\theta} + ((1 - u_1)(1 - u_2)^{1+\theta})^{-\theta/(\theta+1)})^{-1/\theta} \tag{S21}$$

### 4.3.3 Survival Clayton copula with respect to first argument

This copula has negative dependence with tail at $u_1$ high, $u_2$ low.
**Cumulative distribution function:**

$$
\begin{aligned}
C(u_1, u_2; \theta) &= u_2 - C_{\text{Clayton}}(1 - u_1, u_2; \theta) & \text{(S22)} \\
&= u_2 - \left( \max \left\{ (1 - u_1)^{-\theta} + u_2^{-\theta} - 1, 0 \right\} \right)^{-1/\theta}
\end{aligned}
$$

**Density:**

$$
\frac{\partial^2 C(u_1, u_2; \theta)}{\partial u_1 \partial u_2} = (1 + \theta)((1 - u_1)u_2)^{-1-\theta} \left( (1 - u_1)^{-\theta} + u_2^{-\theta} - 1 \right)^{-1/\theta - 2} \tag{S23}
$$

**Conditional cumulative distribution function:**

$$
C_{1|2}(u_1|u_2; \theta) = \frac{\partial C(u_1, u_2; \theta)}{\partial u_2} = 1 - \left( \max \left\{ u_2^{-(1+\theta)}((1 - u_1)^{-\theta} + u_2^{-\theta} - 1)^{-(1+1/\theta)}, 0 \right\} \right) \tag{S24}
$$

**Inverse of conditional cumulative distribution function with respect to $u_1$:**

$$
C_{1|2}^{-1}(u_1|u_2; \theta) = 1 - (1 - u_2^{-\theta} + ((1 - u_1)u_2^{1+\theta})^{-\theta/(\theta+1)})^{-1/\theta} \tag{S25}
$$

### 4.3.4 Survival Clayton copula with respect to second argument

This copula has negative dependence with tail at $u_1$ low, $u_2$ high.
**Cumulative distribution function:**

$$
\begin{aligned}
C(u_1, u_2; \theta) &= u_1 - C_{\text{Clayton}}(u_1, 1 - u_2; \theta) & \text{(S26)} \\
&= u_1 - \left( \max \left\{ u_1^{-\theta} + (1 - u_2)^{-\theta} - 1, 0 \right\} \right)^{-1/\theta}
\end{aligned}
$$

**Density:**

$$
\frac{\partial^2 C(u_1, u_2; \theta)}{\partial u_1 \partial u_2} = (1 + \theta)(u_1(1 - u_2))^{-1-\theta} \left( u_1^{-\theta} + (1 - u_2)^{-\theta} - 1 \right)^{-1/\theta - 2} \tag{S27}
$$

**Conditional cumulative distribution function:**

$$
C_{1|2}(u_1|u_2; \theta) = \frac{\partial C(u_1, u_2; \theta)}{\partial u_2} = \left( \max \left\{ (1 - u_2)^{-(1+\theta)}(u_1^{-\theta} + (1 - u_2)^{-\theta} - 1)^{-(1+1/\theta)}, 0 \right\} \right) \tag{S28}
$$

**Inverse of conditional cumulative distribution function with respect to $u_1$:**

$$
C_{1|2}^{-1}(u_1|u_2; \theta) = (1 - (1 - u_2)^{-\theta} + (u_1(1 - u_2)^{1+\theta})^{-\theta/(\theta+1)})^{-1/\theta} \tag{S29}
$$

# 5 Potential dependence of mixed vine-based models on external variables

The mixed vine-based models that we propose are fully parametric models. Therefore, a subset or all parameters of the margins and of the copulas can be made dependent on external variables. For instance, suppose that the model is 10-D and we decide to make margins 1-5 (representing spike counts of neurons in the visual pathway) depend on the orientation of a bar. If we know the tuning curves of the neurons, we can directly plug them into the rate parameter of the margins. If not, we can still fit the parameters of margins 1-5 depending on the orientation. Depending on whether the element with the higher sampling rate carries substantial information, one can either downsample the signal with the higher sampling rate or upsample the signal with the lower sampling rate. The latter can be accomplished by interpolating between samples or by assuming constant missing signals. In the present form, the model does not make any assumptions or use of different sampling rates. Due to its purly statistical nature, it will still pick up any information there is in the different signals.

# 6 VERTEX network parameters

We used the Virtual Electrode Recording Tool for EXtracellular potentials (VERTEX) [7] to simulate biologically realistic network activity. The parameters of this network were taken from VERTEX tutorial 2 [1] except for the mean random currents in the second stimulus condition. Here we list all parameters that we used.

We simulated a cube of tissue with dimensions X=2500 μm, Y=400 μm and Z=200 μm containing 1 layer and 10 strips with an extracellular medium conductivity of 0.3 S/m. In all cases, the spiking dynamics followed an adaptive exponential model [3].

## 6.1 Pyramidal cells

85% of the total cell population represented pyramidal cells. The spike generation threshold was set to $-50$ mV with spike steepness parameter 2 mV, scale factor of the spike after-hyperpolarization (AHP) current 2.6 nS, AHP current time constant 65 ms and instantaneous change in the AHP current after a spike set to 220 pA. The membrane potential that the soma compartment was reset to after firing a spike was set to $-60$ mV. We used $-45$ mV as the membrane potential at which a spike was detected. Each neuron contained 8 compartments with compartmental parent tree structure [0, 1, 2, 2, 4, 1, 6, 6], lengths 13, 48, 124, 145, 137, 40, 143, 143 μm, diameters 29.8, 3.75, 1.91, 2.81, 2.69, 2.62, 1.69, 1.69 μm, X-(start,end) coordinates in microns (0,0), (0,0), (0,124), (0,0), (0,0), (0,0), (0,-139), (0,139), Y-(start,end) coordinates in microns (0,0), (0,0), (0,0), (0,0), (0,0), (0,0), (0,0), (0,0) and Z-(start,end) coordinates in microns (-13,0), (0,48), (48,48), (48,193), (193,33), (-13,-5), (-53,-13), (-53,-13). All dendrites were aligned along the Z-axis.

**Passive properties.** We set the specific membrane capacitance to 2.96 μF/mm², the specific membrane resistance to 20000/2.96 Ω/mm², the axial resistance to 150 Ωcm and the membrane leak conductance reversal potential to $-70$ mV.

**Input.** We simulated input by means of random currents following an Ornstein-Uhlenbeck process with time constant 2 ms and standard deviation 90 pA. We had two stimulus conditions. In condition 1, we

set the mean value of the process to 330 pA and in condition 2 we set the mean value to 300 pA.

## 6.2 Basket interneurons

15% of the total cell population represented basket interneurons. The spike generation threshold was set to $-50$ mV with spike steepness parameter 2 mV, scale factor of the AHP current 0.4 nS, AHP current time constant 10 ms and instantaneous change in the AHP current after a spike set to 40 pA. The membrane potential that the soma compartment was reset to after firing a spike was set to $-65$ mV. Again, we used $-45$ mV as the membrane potential at which a spike was detected. Each neuron contained 7 compartments with compartmental parent tree structure [0, 1, 2, 2, 1, 5, 5], lengths 10, 56, 151, 151, 56, 151, 151 µm, diameters 24, 1.93, 1.95, 1.95, 1.93, 1.95, 1.95 µm, X-(start,end) coordinates in microns (0,0), (0,0), (0,107), (0,-107), (0,0), (0,-107), (0,107), Y-(start,end) coordinates in microns (0,0), (0,0), (0,0), (0,0), (0,0), (0,0), (0,0) and Z-(start,end) coordinates in microns (-10,0), (0,56), (56,163), (56,163), (-10,-66), (-66,-173), (-66,-173).

**Passive properties.** We set the specific membrane capacitance to 2.93 µF/mm$^2$, the specific membrane resistance to 15000/2.93 $\Omega$/mm$^2$, the axial resistance to 150 $\Omega$cm and the membrane leak conductance reversal potential to $-70$ mV.

**Input.** We simulated input by means of random currents following an Ornstein-Uhlenbeck process with time constant 0.8 ms and standard deviation 50 pA. For the interneurons, we set the mean value of the process to 190 pA in stimulus condition 1 and to 40 pA in condition 2.

## 6.3 Connectivity parameters

We used single exponential current based synapses for all connections.

From pyramidal neurons to pyramidal neurons each presynaptic neuron made 1700 connections to postsynaptic neurons with weight 1 pA and an exponential decay time constant set to 2 ms. From pyramidal neurons to basket interneurons each presynaptic neuron made 300 connections to postsynaptic neurons with weight 28 pA and an exponential decay time constant set to 1 ms. The axonal arbor for these connections followed a 2D-Gaussian connection probability profile with standard deviation 250 µm, cutoff 500 µm and sliced synapses.

From basket interneurons to basket interneurons each presynaptic neuron made 1000 connections to postsynaptic neurons with weight $-5$ pA and an exponential decay time constant set to 6 ms. From basket interneurons to pyramidal neurons each presynaptic neuron made 200 connections to postsynaptic neurons with weight $-4$ pA and an exponential decay time constant set to 3 ms. The axonal arbor for these connections followed a 2D-Gaussian connection probability profile with standard deviation 200 µm, cutoff 500 µm and sliced synapses.

In all cases, axonal conduction speed was set to 0.3 m/s and the synapse release delay was set to 0.5 ms.

## Footnotes

[1]We could also extend the range of the Clayton parameter $\theta$ to include values in $[-1, 0)$. This would yield negative Clayton dependence but with a shape different from that for positive dependence. Instead, we use survival Clayton copulas to model negative dependence in a form that has the same shape as for positive dependence.