[Reviews · NeurIPS 2016]

Reviewer 1

Summary

The authors present a vine copula model, which is suitable for high dimensional data including mixed observation types (continuous and discrete). The given model can be fit to data and sampling-based estimates can be used to compute information theoretic quantities on the data. This method appears to be both tractable and of interest to those wishing to model data from complex datasets that include distinct types of observations, such as neuron spike counts and continuous calcium signals.

Qualitative Assessment

A general concern for this type of model for neuroscientific applications is that external variables, such as a visual stimulus, are not explicitly included. Is it complicated or intractable to place external variables like a visual stimulus or motor output into the model? Additionally, while the model can handle different forms of observations, it may be very difficult to figure out how to format data for this type of models: continuous calcium signals have a significantly lower sampling rate than intracellular voltage signals.

Confidence in this Review

1-Less confident (might not have understood significant parts)


Reviewer 2

Summary

The authors develop tractable means of sampling from and calculating the likelihood of C-vine copulas which permit both discrete and continuous marginal distributions. The proposed algorithms for sampling and likelihood calculation are extension of previous work (that of Aas et al. for likelihood calculation and Panagiotelis et al. for sampling). The benefit of the methods proposed is that while naive sampling strategies for calculating likelihoods and sampling would scale exponentially with the number of discrete random variables in a model, the proposed methods scaled quadratically. The authors propose to apply such copulas to the study of neural data which consists of both continuous measurements (such as local field potentials) and discrete measurements (spike counts). The authors demonstrate that data generated from such mixed copulas is better fit by these models than by models which assume independence between the marginal variables or use continuous distributions to model variables which are in fact discrete. They also simulate LFP and spike data from a network driven by two different stimulus conditions and show that the mutual information between the stimulus and recorded neural signals differs substantially when it is calculated with the mixed copulas proposed in this work vs. a model which assume independence between marginal variables or a copula model which assume all marginal variables as continuous.

Qualitative Assessment

The development of flexible methods to model the joint distribution between continuous and random variables is a important problem with many application areas, one of which, as the authors note, is neuroscience. Copula which allow for both discrete and continuous random variables are one means of approaching this problem, and the development of general and computationally tractable methods for fitting and performing inference with such models is of broad interest. The paper makes multiple methodological contributions, which I find valuable. The proposed family of models seems flexible and likely useful in practice. While others have previously proposed pair copula constructions as well as efficient algorithms for sampling from discrete copulas, the development of pair copula constructions and associated efficient algorithms for sampling and inference for mixed discrete and continuous data is valuable. However, I feel that other aspects of the paper could be improved. I believe the paper would benefit from more clarity. There were multiple parts of the paper that I found difficult to follow. For example, equation 2 provides a general formula for the likelihood of data under a mixed copula model, but it was not until after reading some of the cited work that I was able to understand it was derived by taking differences of the discrete random variables. While I was previously familiar with copula in general, this was my first exposure to C-vine copula. The description of C-vines required additional reading for me to follow. It is certainly understandable that it may be hard to fully explain all concepts in an 8 page paper, but I believe a general NIPS audience would benefit from the provision of a little more explanation and background. There were also symbols which were used without introduction (the super script +/- signs which I believe indicate differences, "c" for the pdf of a copula, etc.). While it was possible for me to follow the gist of paper, it was difficult to fully follow all the details. It would be helpful to explain why a copula approach might be beneficial over existing approaches which have been applied in the neuroscience literature. While it is not referenced in the present paper, there is established body of work in the neuroscience community which has examined the relationship between neural spiking and LFP (e.g., the first three references below and references therein). Finally, I believe the results section would benefit from more depth. If I understood correctly, in the first analysis data was generated from a particular mixed copula model. The fit of mixed copula models, models which assumed independent random variable and copula models which modeled all marginal variables as continuous to this data was then compared. Mixed copula models were shown to best fit the data. While this is reassuring, this result also seems expected, as models which are subject to mismatch would not be expected to fit data as well as a model which is of the same family the data was generated from. To show the utility of proposed methods, an application to more realistic data is beneficial, and the author's provide this by applying their methods to simulated spike and LFP data and show that mutual information (MI) between a stimulus and the simulated LFP and spike data is different under mixed copula models vs. models which assume independent random variables or require all random variables to be continuous. I find the author's claim that a mixed copula model vs. the other two options should best fit the data believable, and seeing the difference in the values of MI between the calculated models is interesting. However, if an actual application scenario, I believe experimentalists would not be primarily concerned with the value of MI returned - but by differences in MI returned by the same model in different experimental contexts. For example, one might ask if MI between LFP and neurons in two areas changes when a subject is attending vs. not attending to a stimulus. In such a scenario the primary question of interest is if a statistical method is sensitive enough to reveal differences between MI between the two conditions, and thus it would perhaps be more interesting to see an analysis showing the benefit of the proposed methods in such a setting. References: Lepage, K. Q., Gregoriou, G. G., Kramer, M. A., Aoi, M., Gotts, S. J., Eden, U. T., & Desimone, R. (2013). A procedure for testing across-condition rhythmic spike-field association change. Journal of neuroscience methods, 213(1), 43-62. Kelly, R. C., Smith, M. A., Kass, R. E., & Lee, T. S. (2010). Local field potentials indicate network state and account for neuronal response variability. Journal of computational neuroscience, 29(3), 567-579. Carlson, D. E., Borg, J. S., Dzirasa, K., & Carin, L. (2014). On the relations of LFPs & Neural Spike Trains. In Advances in Neural Information Processing Systems (pp. 2060-2068). K. Aas, C. Czado, A. Frigessi, and H. Bakken. Pair-copula constructions of multiple dependence. Insurance: Mathematics and Economics, 44(2):182–198, 2009. A. Panagiotelis, C. Czado, and H. Joe. Pair copula constructions for multivariate discrete data. Journal of the American Statistical Association, 107(499):1063–1072, 2012.

Confidence in this Review

2-Confident (read it all; understood it all reasonably well)


Reviewer 3

Summary

The paper presents a copula-based family of probabilistic models for data with mixed discrete and continuous variables. a curse of dimensionality problem is avoided limiting the structure of the interactions of the discrete variables. The usefulness of the models is shown on simulated LFP and spike data from a simulated neuronal network. It is shown that model likelihood estimates and mutual information estimates are greatly affected by choice of model when comparing the mixed copula models versus independent-variables models.

Qualitative Assessment

The paper is well written though the description of C-vine copulas is very hard to understand without the supplementary material and references. The simulations end experiment on simulated neuronal data are well thought out and convincing of the usefulness of the method. I did not understand the construction if C-vine copulas. The supplementary was a bit helpful but not enough for me. The combination of discrete and continuous variables seems very straight-forward. As a paper that addresses all the issues that need be solved for using such models with mixed neural data, I find it quite important and a significant advance. (Though my personal feeling is that until a full length journal paper is published, it will only find an audience within a select group who are already familiar with all the C-vine-related modelling and have an interest in modelling mixed data).

Confidence in this Review

1-Less confident (might not have understood significant parts)


Reviewer 4

Summary

Copula is a canonical method to describe dependency structure of discrete and continuous random variables by separating it from description of marginal distributions. While there are parametric copulas for multivariate random variables such as Frank or Gaussian copulas, these models may be limit. Here the author(s) proposed a method for constructing a joint model by iteratively combining bivariate copula models. Sampling method from the model and fitting procedure were proposed based on known procedures. The author(s) verified their model to simulated mixture signals of count and continuous data. They then applied their method to simulated LFP and spike count data using a virtual electrode recording toolbox, and demonstrated that mutual information between neural activity and stimulus conditions is significantly influenced by the dependency modeling.

Qualitative Assessment

Recently several authors started to apply copula to model mixture signals of count and continuous data for neuroscience data. This manuscript provides a practical method that is scalable to analysis of larger number of neurons and LFPs, by iteratively applying pairwise analyses. Although the descriptions are complicated, the underlying idea is straightforward, thus was easy to grasp. It may be helpful if the author(s) could provide an intuitive picture of the tree structure described at lines 105-109. Below I describe questions that arose by reading this manuscript. I hope that these questions help the author(s) to extend their manuscript. * This method is based on pairwise analysis, thus likely to neglect higher-order dependency among signals. If so, then how much do we miss the higher-order depndency? If this is difficult to assess, it would be better if author(s) could comment on this issue in the manuscript. * Gaussian copula is probably the easiest model to construct multivariate dependency. Thus it would be nicer if the authors could compare performance of the vine copula model with a Gaussian copula model as these share the nature of pairwise analysis. * The final result on the mutual information (MI) shows reduction of MI by constraining the dependency among the signals. Does this mean that count and LFP signals are redundantly code the two stimulus conditions? It is recommended that the author(s) extend implications of this result.

Confidence in this Review

2-Confident (read it all; understood it all reasonably well)


Reviewer 5

Summary

In this paper, the authors suggested a theoretical framework, which is based on vine copulas, to characterize stochastic models with mixed (continuous and discrete) variables. They demonstrated how such a framework can be used to generate samples, to calculate probability densities and to infer parameters based on empirical observed samples. Further, this method was used to analyze simulated neuronal dynamics in order to extract information jointly encoded by local field potentials (the continuous variables) and the spiking activities (the discrete variables).

Qualitative Assessment

By extending vine copulas to deal with mixed variables, this paper presents a potentially useful method to analyze local field potentials and spiking activities jointly. Technical quality: Overall it seems all right to me. However, a few things could be further improved. In figure 2, it would be informative to show the results of the best fitting mixed vine based model (besides the ground true model). In figure 3, it seems that a classifier based on the independent model would do much better. Why? Did the vine-based models introduce spurious information or it is simply because with limited sample size the parameters of more complex models cannot be estimated accurately? Novelty: It is a novel approach, obtained through relatively straightforward extension of previously introduced ones. Potential usefulness: I have two concerns: 1. How feasible to infer parameters accurately for high dimensional systems? 2. The somewhat controversial results shown in Fig.3 B. If an independent model can work better in terms of extracting information, how useful the current method would be for practical application, e.g., brain-machine interface? Clarity of the presentation: It might be helpful to explain more clearly why it is important to combine LFP and spikes, i.e., what extra information it can provide beyond the conventional method?

Confidence in this Review

2-Confident (read it all; understood it all reasonably well)